# Comparison of longitudinal trends in self-reported symptoms and COVID-19 case activity in Ontario, Canada

**Arjuna S. Maharaj** [1]*, **Jennifer Parker** [1], **Jessica P. Hopkins** [2,3,4], **Effie Gournis** [4,5], **Isaac I. Bogoch** [6,7], **Benjamin Rader** [8,9], **Christina M. Astley** [8,10,11], **Noah M. Ivers** [12,13], **Jared B. Hawkins** [8], **Liza Lee** [14], **Ashleigh R. Tuite** [4], **David N. Fisman** [4,6], **John S. Brownstein** [8,15], **Lauren Lapointe-Shaw** [6,7]

1 Doctor of Medicine Program, Temerty Faculty of Medicine, University of Toronto, Toronto, Canada, 2 Public Health Ontario, Toronto, Canada, 3 Department of Health Research Methods, Evidence, and Impact, McMaster University, Hamilton, Canada, 4 Dalla Lana School of Public Health, University of Toronto, Toronto, Canada, 5 Toronto Public Health, City of Toronto, Toronto, Canada, 6 Department of Medicine, University of Toronto, Toronto, Canada, 7 Department of Medicine, University Health Network, Toronto, Canada, 8 Computational Epidemiology Lab, Boston Children's Hospital, Boston, MA, United States of America, 9 Department of Epidemiology, Boston University, Boston, MA, United States of America, 10 Division of Endocrinology, Harvard Medical School, Boston Children's Hospital, Boston, MA, United States of America, 11 Broad Institute of Harvard and MIT, Cambridge, MA, United States of America, 12 Women's College Research Institute, Toronto, Canada, 13 Department of Family and Community Medicine, University of Toronto, Toronto, Canada, 14 Centre for Immunization and Respiratory Infectious Diseases, Public Health Agency of Canada, Ottawa, ON, Canada, 15 Department of Pediatrics and Biomedical Informatics, Harvard Medical School, Boston, MA, United States of America

* a.maharaj@mail.utoronto.ca

## Abstract

### Background

Limitations in laboratory diagnostic capacity impact population surveillance of COVID-19. It is currently unknown whether participatory surveillance tools for COVID-19 correspond to government-reported case trends longitudinally and if it can be used as an adjunct to laboratory testing. The primary objective of this study was to determine whether self-reported COVID-19-like illness reflected laboratory-confirmed COVID-19 case trends in Ontario Canada.

### Methods

We retrospectively analyzed longitudinal self-reported symptoms data collected using an online tool–Outbreaks Near Me (ONM)–from April 20th, 2020, to March 7th, 2021 in Ontario, Canada. We measured the correlation between COVID-like illness among respondents and the weekly number of PCR-confirmed COVID-19 cases and provincial test positivity. We explored contemporaneous changes in other respiratory viruses, as well as the demographic characteristics of respondents to provide context for our findings.

### Results

Between 3,849–11,185 individuals responded to the symptom survey each week. No correlations were seen been self-reported CLI and either cases or test positivity. Strong positive

**Data Availability Statement:** We have attached a supplemental data file which contains aggregate data for all figures and major conclusions in our manuscript. We are unable to provide individual-

level data for our data sources as they are third-party and not owned by us. We have provided a description of each data set used and any contact information other would need to apply to gain access to the data as asked. Outbreaks Near Me: Outbreaks Near Me is a third-party data set owned by researchers at Boston Children's Hospital containing the self-reported symptom data on one of the participatory surveillance tools used in the manuscript. More information can be obtained from: https://outbreaksnearme.org/us/en-US/ Requests for access can be made to John. Brownstein@childrens.harvard.edu. FluWatchers: FluWatchers is a third-party data is owned by the Public Health Agency of Canada. It contains self-reported cases of influenza-like illness. It can be accessed at: https://www.canada.ca/en/public-health/services/diseases/flu-influenza/influenza-surveillance/weekly-influenza-reports.html. Requests for access can be made to: liza. lee@canada.ca Regional Respiratory Data: Regional seasonal respiratory virus data is a third-party data set owned by Public Health Ontario. It contains weekly percent positivity of common seasonal respiratory viruses tests for in Ontario. It can be found at the following link: https://www. publichealthontario.ca/en/data-and-analysis/ infectious-disease/respiratory-pathogens-weekly. Notably, for FluWatchers and seasonal respiratory data, we only had access to the aggregated results needed to produce our graphs and test for correlation. COVID-19 Confirmed Case Reports and Testing Data: Confirmed COVID-19 cases in Ontario are collected by Public Health Ontario and stored in Ontario's Case and Contact Management Plus. This is a third-party data source owned and can be accessed at: https://data.ontario.ca/en/ dataset/covid-19-vaccine-data-in-ontario. Laboratory SARS-CoV2 testing data is owned by the Ontario Laboratory Information System (OLIS) and eHealth Ontario. Access to data can be found here: https://ehealthontario.on.ca/en/news/view/ online-access-to-covid-19-lab-test-results-for-health-care-providers 2016 Canadian Census Data: Canadian 2016 census data is owned by Statistics Canada with access provided through the University of Toronto Faculty of Arts & Sciences. Data was access through http://www.chass. utoronto.ca.

**Funding:** This study is funded by the UofT COVID action initiative (LL-S), URL: https://medicine. utoronto.ca/eligibility-update-toronto-covid-19-action-initiative Outbreaks Near Me is funded by the Center for Disease Control and Prevention, Ending Pandemics and Flu Lab. CDC: https://www. cdc.gov Ending Pandemics: https:// endingpandemics.org Flu Lab: https://theflulab.org

correlations were seen between CLI and both cases and test positivity before a previously documented rise in rhinovirus/enterovirus in fall 2020. Compared to participatory surveillance respondents, a higher proportion of COVID-19 cases in Ontario consistently came from low-income, racialized and immigrant areas of the province- these groups were less well represented among survey respondents.

## Interpretation

Although digital surveillance systems are low-cost tools that have been useful to signal the onset of viral outbreaks, in this longitudinal comparison of self-reported COVID-like illness to Ontario COVID-19 case data we did not find this to be the case. Seasonal respiratory virus transmission and population coverage may explain this discrepancy.

## Introduction

Viral surveillance can help detect COVID-19 outbreaks, mobilize a rapid response and thereby reduce morbidity and mortality [1, 2]. However, there are limitations to relying solely on laboratory testing for COVID-19 surveillance. At an individual-level, delays between symptom onset and testing, and between testing and COVID-19 test results mean that reported cases typically reflect disease activity from 1–2 weeks prior [3]. When case counts are high, testing restrictions may be implemented to preserve capacity, amplifying the underestimation of case activity. Typically, restrictions have included prioritizing those with the highest pre-test probability for a positive result (e.g., symptomatic individuals and/or potential exposure to a confirmed case) or those at risk of severe illness [4]. Surveys from the first wave of the COVID-19 pandemic estimated that only 2–9% of Canadians with symptoms consistent with COVID-19 received viral tests [5]. When viral transmission and new case counts are high, further delays in testing and results may reduce the reliability of confirmed case data for identifying key epidemiological events such as exponential growth or curve flattening. These limitations highlight the need for more timely, comparable, and comprehensive methods of population disease surveillance to inform public health measures.

Syndromic surveillance is a public health tool used extensively to identify the beginning of seasonal influenza outbreaks in the United States and Canada, and for the surveillance of other viral and bacterial diseases globally [6–9]. Participatory surveillance, a subtype of syndromic surveillance, allows individuals to self-report symptoms through phone or internet-based applications [10]. Where testing is incomplete, participatory surveillance data for COVID-19 can be used as an adjunct for confirmed case counts to help to estimate the true burden of disease, and forecast future epidemiological trends with strong spatial and temporal resolution [11–13]. There has been increasing global utilization of crowdsourced data for disease surveillance and estimating effectiveness of public health interventions [11–15]. We previously reported a divergence between self-reported symptoms and COVID-19 case numbers in the context of a seasonal peak of rhinovirus/enterovirus, in Ontario, Canada, in fall 2020 [16]. Throughout the three waves of COVID-19 in Ontario, the burden of illness has disproportionately been borne by lower income and marginalized groups [17]. Considering these changes, we first aimed to examine whether Ontario-wide self-reported COVID-19 symptoms were correlated with laboratory-confirmed COVID-19 case trends in 2020–2021. Second, to help interpret the findings, we compared the changing sociodemographic characteristics of

FluWatchers is funded by the Public Health Agency of Canada. PHAC: https://www.canada.ca/en/public-health.html The funders had no role in study design, data collection and analysis, decision to publish, or preparation of the manuscript.

**Competing interests:** We have read the journal's policy and the authors of this manuscript have the following competing interests: IIB has consulted to BlueDot, a social benefit corporation that tracks the spread of emerging infectious diseases. DNF reports personal consultant fees from Pfizer, AstraZeneca, and Seqirus, outside the submitted work. This does not alter our adherence to PLOS ONE policies on sharing data and materials

Ontario's COVID-19 cases to the sociodemographic characteristics of participatory surveillance respondents.

## Overview and setting

We retrospectively analyzed self-reported participatory surveillance COVID-19 symptoms and test results, in addition to laboratory-confirmed COVID-19 case and testing data from Ontario, Canada. Ontario is Canada's most populous province, with approximately 14.5 million residents. The first case of COVID-19 in Ontario was reported on Jan. 25th, 2020, and community transmission was estimated to have started on March 17th, 2020. As of June 2021, the province has experienced three waves of COVID-19. The first wave peaked in mid-April 2020 at a weekly average of approximately 600 new daily cases, although it is believed that cases were considerably undercounted at the time due to restrictive testing policies. The second wave peaked in early-January 2021 at weekly average of approximately 3600 new daily cases. The third wave peaked in mid-April 2021 and low case levels have been achieved as of early June 2021 signalling the wave is likely over.

## Methods

This study was approved by the Ethics Review Board of University Health Network and the University of Toronto, and a waiver of informed consent was granted because the data were collected for public health surveillance purposes. All methods were performed in accordance with institutional guidelines and regulations.

### Data sources and study population

The five data sources used for this study include: 1) participatory surveillance survey data from Outbreaks Near Me (ONM, formerly COVID Near You) and FluWatchers, 2) regional COVID-19 laboratory confirmed case reports from the Ontario Case and Contact Management Plus (CCM Plus), 3) regional laboratory SARS-CoV2 testing data from the Ontario Laboratory Information System (OLIS), 4) 2016 Canadian Census data and 5) Ontario Respiratory Virus Data from the Ontario Respiratory Pathogen Bulletin. We created weekly tabulations of syndromic survey data, COVID-19 case counts and laboratory tests using the International Organization for Standardization (ISO) week (Monday through Sunday) [18].

Outbreaks Near Me (outbreaksnearme.org) is a web-based participatory health surveillance tool created by infectious disease epidemiologists at Boston Children's Hospital and launched in March 2020. This team also created Flu Near You (flunearyou.org), a similar tool for influenza symptoms, which has been validated against clinical data sources and applied to predict influenza trends [6–8]. Participants are asked to report on present symptoms, date of symptom onset, demographic information, area of residence (first three digits of postal code), healthcare encounters, testing, and results. Respondents reported symptoms on the ONM website and could opt to leave their cell phone number to receive SMS reminders to complete the survey again every three days after their initial submission. Overall, 96.0% of responses to ONM in Ontario came from SMS reminders (weekly mean: 96.1%; SD: 5.9%). The mean number of Ontario weekly responses to the ONM SMS survey prompts was 11,289 (mean response rate: 36.2%; SD: 2.0%). Symptoms of possible COVID-19 were defined using the CDC Surveillance Case Definition for COVID-19 from the National Notifiable Diseases Surveillance System (NNDSS). We used the definition of COVID-like illness (CLI) in effect since August 5th, 2020, defined by the presence of at least two of: fever (measured or subjective), chills, rigors, myalgia, headache, sore throat, nausea or vomiting, diarrhea, fatigue, congestion or runny nose or at least one of: cough, shortness of breath, difficulty breathing, new olfactory disorder, or new

taste disorder [19]. This case definition had a reported sensitivity of 97–98% and a specificity of 33–43% in adults for detecting a COVID-19 diagnosis [20]. We identified repeat responses by age/sex/phone number and included only one response per person-week, prioritizing a CLI positive response and, if none occurred, the first response in each week. We included responses with a self-reported postal code originating from Ontario, Canada, between April 20[th], 2020 (week 17) and March 7[th], 2021 (week 9).

FluWatchers (https://www.canada.ca/en/public-health/services/diseases/flu-influenza/influenza-surveillance/weekly-influenza-reports.html) is an internet-based participatory surveillance tool created by the Public Health Agency of Canada in November 2015 to track Influenza-like Illness (ILI). Defined as the presence of fever and cough, ILI has a reported sensitivity of 51–54% and specificity of 86–90% for a COVID-19 diagnosis in adults [21]. Participants can sign up to receive weekly email reminders to report symptoms through a link to an online platform. A total of 9,756 users reported symptoms at least once between April 20[th], 2020 and March 7[th], 2021 in Ontario, and among these users, the average weekly response rate between weeks 17 of 2020 and week 9 of 2021 was 68% (range 60–88%).

CCM Plus data system has been implemented in Ontario to record COVID-19 case information. Each of Ontario's 34 public health units is responsible for local COVID-19 case investigation and entry of case information into CCM Plus. We obtained confirmed COVID-19 case counts from the CCM Plus data system on March 12[th], 2021 for the time period between April 20[th], 2020 and March 7[th], 2021. Extracted de-identified data included case reported date, accurate episode date (date of symptom onset, or if not present the date of specimen collection), age, gender, symptomatic status, and area of residence (first three digits of postal code). We used the accurate episode date to estimate the date of symptom onset. We extracted a separate dataset from Ontario Laboratory Information System of the total daily COVID-19 tests by age, gender and area of residence, with data ranging from April 20[th], 2020 to March 7[th], 2021. Weekly percent positivity in Ontario was calculated by dividing total positive cases reported each week by the total number of tests reported each week.

The Canadian Census collects information through survey of individuals across Canada on their demographic, social and economic factors [22, 23]. Data were obtained for all forward sortation areas (FSA; designated geographical unit based on the first three characters in a Canadian postal code) in Ontario. We obtained median household income, percent recent immigrants (those immigrating in the last 5 years), and percent visible minority, by FSA, from the 2016 census. Based on each of these variables, we divided the Ontario's 523 FSAs into 5 quintile groups. We then assigned each ONM respondent, COVID-19 case and individual tested the three sociodemographic variables based on their reported FSA of residence. We then plotted trends in these variables for both ONM respondents and COVID-19 cases in Ontario over time by the five Ontario quintile groups. FSAs also contain information on an individual's area of dwelling (urban or rural) in the second digit [24]. This was used to calculate and compare the proportion of survey respondents living in urban and rural areas to that of the Ontario general population, those tested and laboratory-confirmed cases of COVID-19.

Data on the percent positivity of non-SARS-CoV2 respiratory pathogens were obtained from the Ontario Respiratory Pathogen Bulletin (ORPB). This provides a weekly summary of the laboratory-confirmed percent positivity of eight common respiratory viruses in Ontario. These data are submitted to the Public Health Agency of Canada from 16 participating laboratories in Ontario, including 11 Public Health Ontario Laboratories and five hospital-based laboratories. Data were extracted on March 8[th], 2021. Test positivity of the eight common respiratory viruses were plotted from April 20[th], 2020 –March 7[th], 2021.

## Analysis

**Syndromic trends.** To assess the relationship between CLI from ONM and COVID-19 activity in Ontario, we compared both the weekly percent positivity in Ontario and the weekly number of new reported cases against the proportion of ONM respondents reporting CLI a) one week prior and b) the same week. We used both contemporaneous and one-week future indicators because of the potential for participatory surveillance to anticipate provincial COVID-19 case data, particularly in light of the known delays between symptom onset and positive case reporting. We also compared participatory surveillance data to COVID-19 case activity in the weeks before, during and after a provincial rise in other seasonal respiratory viruses previously documented [16]. For each of these, we reported Spearman's rank correlation coefficient, and determined statistical significance using a t-test. Clopper-Pearson confidence intervals were calculated and plotted as error bars for all proportions. The data were analyzed using R version 4.0.1 in the RStudio software environment, version 1.1.463 (RStudio Inc., Boston, MA). All testing for differences was done at a two-tailed $p <0.05$ significance threshold.

**Sensitivity analyses.** We conducted four sensitivity analyses to confirm our findings. We compared cases in Ontario to two alternative syndromic definitions. The first alternative definition ($CLI_2$) consisted of cough or fever or loss of smell or taste. These three symptoms had the strongest predictive value of self-reported COVID-19 test positivity across three national digital surveillance platforms [25]. The second alternative syndromic definition ($CLI_3$) consisted of taste and/or smell dysfunction, or any one of: shortness of breath, myalgia, fever, or chills. This definition had a reported 95% specificity and 76% sensitivity for laboratory confirmed SARS-CoV2 [20]. A syndromic definition with high specificity was chosen in order to be less likely affected by other respiratory viruses (e.g. Rhinovirus or enteroviruses) [26]. Next, we compared the proportion of ONM respondents reporting CLI based on the week of symptom onset to the number of cases in Ontario based on the accurate episode date (a proxy for symptom onset date). After that, we restricted the comparison to provincial COVID-19 cases that were symptomatic, as asymptomatic testing practices varied over time, and asymptomatic cases would not be detected through participatory surveillance. In addition, because children have been described more commonly to have asymptomatic COVID-19 infection, we restricted both CLI and COVID-19 confirmed cases to those aged 19 years and older and repeated the comparison [27]. Finally, in a post-hoc analysis suggested at peer review we compared weekly COVID-19 cases to the weekly percentage of respondents reporting close contact with a confirmed SARS-CoV2 case, and close contact with CLI symptoms.

**Comparison across syndromic data sources.** To compare the rate of syndromic signal across differing participatory surveillance platforms, we compared the weekly proportion of ONM respondents with ILI to the weekly proportion of FluWatchers respondents with ILI.

**Demographics.** We compared ONM respondent characteristics to those of the general Ontario population, those undergoing COVID-19 testing and laboratory-confirmed COVID-19 cases in Ontario. Provincial population estimates on July 1st, 2020, by age and sex, were obtained from Statistics Canada [28]. Testing for differences in proportions was done using Chi-square tests and Fisher exact tests (if small cells). The age distributions of those reporting CLI and positive COVID-19 cases were plotted by week.

## Results

### Outbreaks Near Me respondents, April 20—March 7th, 2021

There were 525,014 total responses from 67,693 unique respondents to the ONM survey between April 20th, 2020 and March 7th, 2021. After removing duplicate respondents from

each week, 297,246 responses were identified for analysis. The total number of unique responses per week ranged from 3,849–11,185 with a mean of 6,461 weekly responses with relative stability over time (Fig 1 in S1 Appendix).

## Outbreaks Near Me symptom and CLI reporting

Overall, CLI was reported in 1.40% (n = 4,147) of responses, while 1.62% (n = 4,819) of all responses reported at least one symptom. The most commonly reported CLI symptom was fatigue (n = 2,290; 0.77%) and the least reported CLI symptom was loss of smell or taste (n = 267; 0.09%) (Fig 2 in S1 Appendix). There were two observable rises in CLI, with the first occurring in week 20 (May 11th–May 18th, 2020) and the second occurring in week 41 (October 5th, 2020). In the first rise, the top three components of CLI included fatigue, headache, and congestion or runny nose while in the second rise, the top three components of CLI included sore throat, congestion or runny nose and fatigue (Fig 3 in S1 Appendix).

## Comparison of survey and SARS-CoV-2 data

**Same week.**   There was no correlation between the weekly number of reported cases in Ontario and CLI each week ($r_s$ = 0.02, $p$ = 0.91, Fig 1A) and no correlation between test percent positivity in Ontario and CLI ($r_s$ = 0.09, $p$ = 0.56, Fig 1B) over the entire time period. No correlation was also seen between CLI and symptomatic COVID-19 cases over the entire time period ($r_s$ = 0.01, $p$ = 0.94, Fig 1A). Strong positive and significant correlations were seen only in the weeks before the rise in rhino/enterovirus positivity in fall 2020 (Table 1 in S1 Appendix). A large increase in enterovirus/rhinovirus percent positivity was seen in Ontario starting in August 2020 (week 34), peaking in September 2020, and gradually falling into January 2021. Enterovirus/rhinovirus levels returned to baseline levels at week 2 of 2021 (Fig 2).

**One-week future cases.**   After incorporating a one-week lag by comparing self-reported symptoms to test results in the following week, there was similarly no correlation between self-reported CLI and either reported case numbers or percent positivity (Table 1 in S1 Appendix). In contrast, strong positive correlations were seen in each of these analyses prior to the rise of rhinovirus activity in fall 2020 (Table 1 in S1 Appendix).

**Sensitivity analyses.**   Using the alternative $CLI_2$ and $CLI_3$ syndromic definitions did not meaningfully change the results (Table 1 in S1 Appendix). Substituting estimated symptom onset date for reported date in laboratory-confirmed cases and survey responses, restricting the comparison to symptomatic COVID-19 cases, and restricting the comparison to those aged 19 years and above also did not meaningfully change the results (Table 1 in S1 Appendix). A strong positive correlation was also seen between weekly cases and those self-reporting CLI symptoms and direct contact with a confirmed case ($\rho$ = 0.70), however this was notably less than the correlation between cases and reported close contacts alone ($\rho$ = 0.77, Fig 6 in S1 Appendix).

## Comparison across syndromic data sources

The proportion of ONM respondents reporting ILI (fever and cough) each week ranged from a high of 0.21% (n = 13) in week 39 (Sept. 21st– 27th) to a low of 0% (n = 0) in week 5, 2021 (Feb 1st–Feb 7th). The proportion of respondents reporting ILI from ONM and from Flu-Watchers had similar ranges and trends over time (Fig 4 in S1 Appendix). There was a moderate positive correlation in the weekly percentage of respondents reporting ILI between the ONM and FluWatchers survey ($r_s$ = 0.52, $p$ < 0.01).

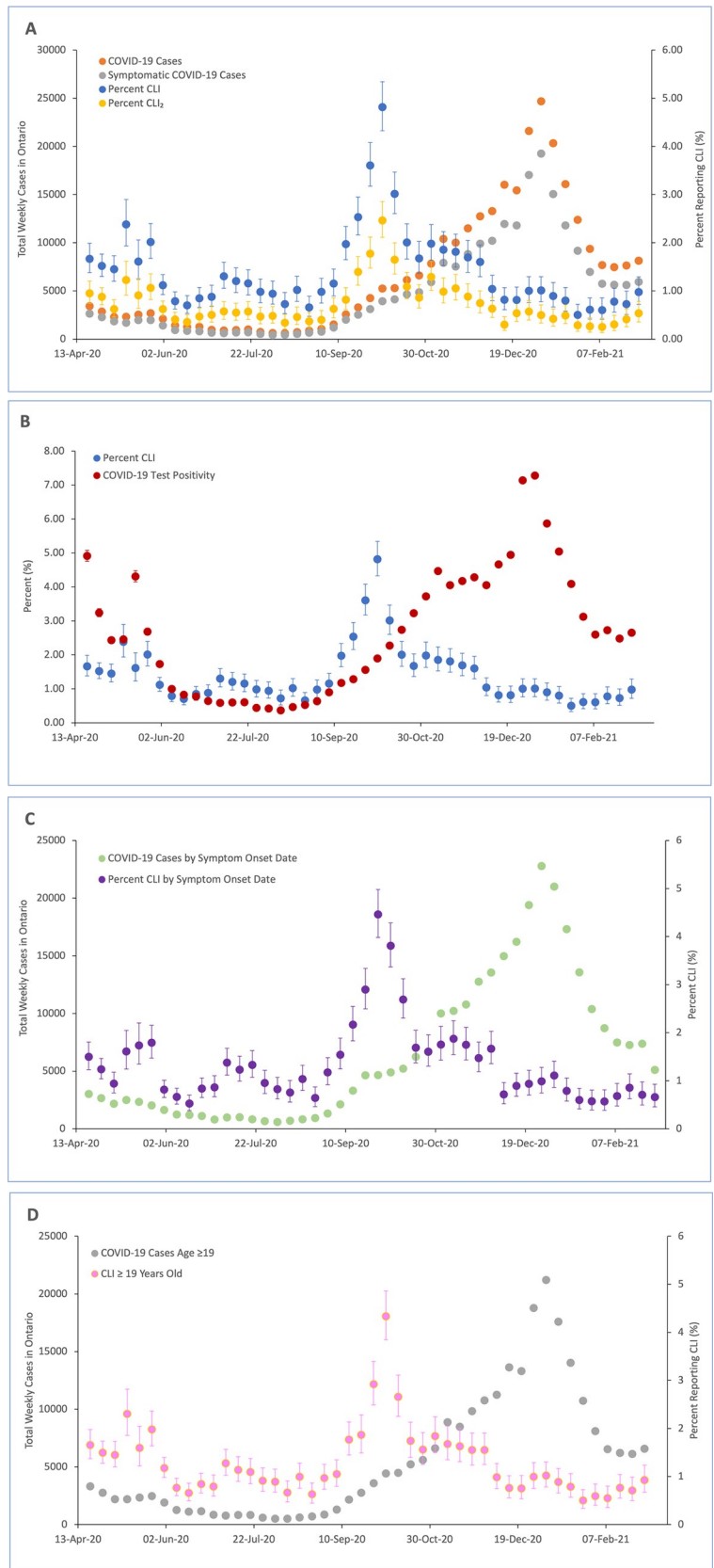

**Fig 1. Comparison of surveillance signal from ONM to COVID-19 activity.** (A) Percent CLI and CLI$_2$ vs new COVID-19 cases and symptomatic COVID-19 cases. (B) Percent CLI and percent positivity for SARS-CoV2. (C) Percent CLI and number of new COVID-19 cases based on the estimated date of symptom onset. (D) Percent CLI of those ≥19 years of age and new COVID-19 cases among those ≥19 years of age.

### Sociodemographic characteristics overall and over time

**Age.**  The proportion of ONM respondents aged 40–59 years (n = 29,206; 43.1%) was significantly higher than that of the tested population (n = 3,141,700; 31.8%, $p < 0.01$) and the Ontario population overall (n = 3,915,662; 26.9%, $p < 0.01$). There was also a significantly smaller portion of respondents who were <19 years old in ONM (n = 3,072; 4.5%) compared to those who received a test (n = 1,020,528; 10.3%, $p < 0.01$) and the Ontario general population (n = 3,141,693; 21.6%, $p < 0.01$). The age distribution of ONM respondents did not change over time. The <19 years age demographic consistently made up the lowest proportion of respondents, while the 40–59 age demographic was consistently the most likely to respond each week (Fig 3).

There was an increasing proportion of younger people (≤39 years) reporting CLI form April–October 2020. In April 2020, approximately 30% of those reporting CLI were ≤39. This steadily increased to ~ 60% in October 2020. A similar trend was seen in COVID-19 cases in Ontario with those ≤39 increasing from ~25%– 60% between the period of April– October 2020. However, there was a subsequent decrease in those ≤39 reporting CLI after October 2020. This trend was not observed in COVID-19 cases as the proportion of those ≤39 remained elevated and stable at ~50% with an increase in March 2021 (Fig 4).

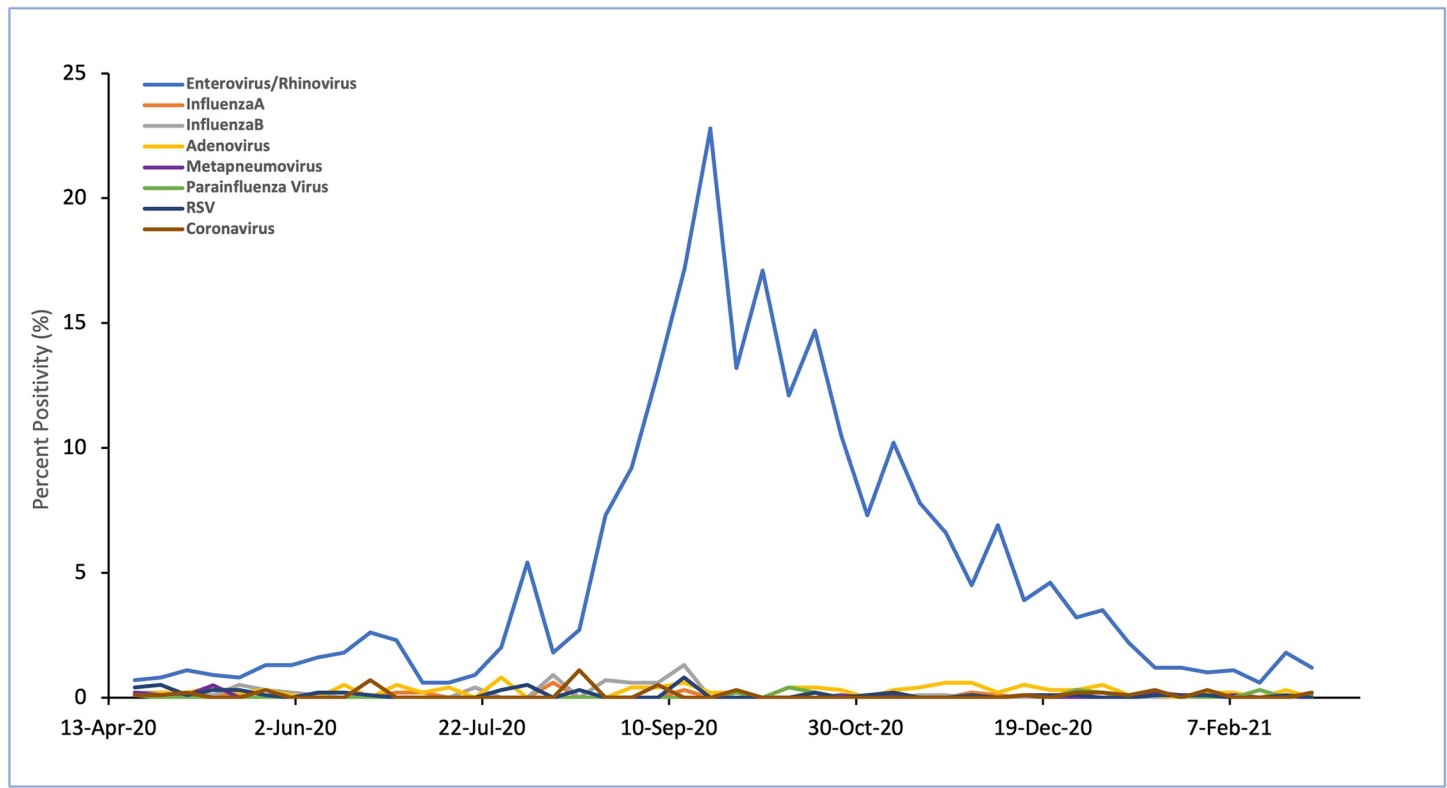

**Fig 2. Percent positivity of seasonal respiratory viruses.** Coronavirus represents tests positivity of non-SARS-CoV2 coronaviruses.

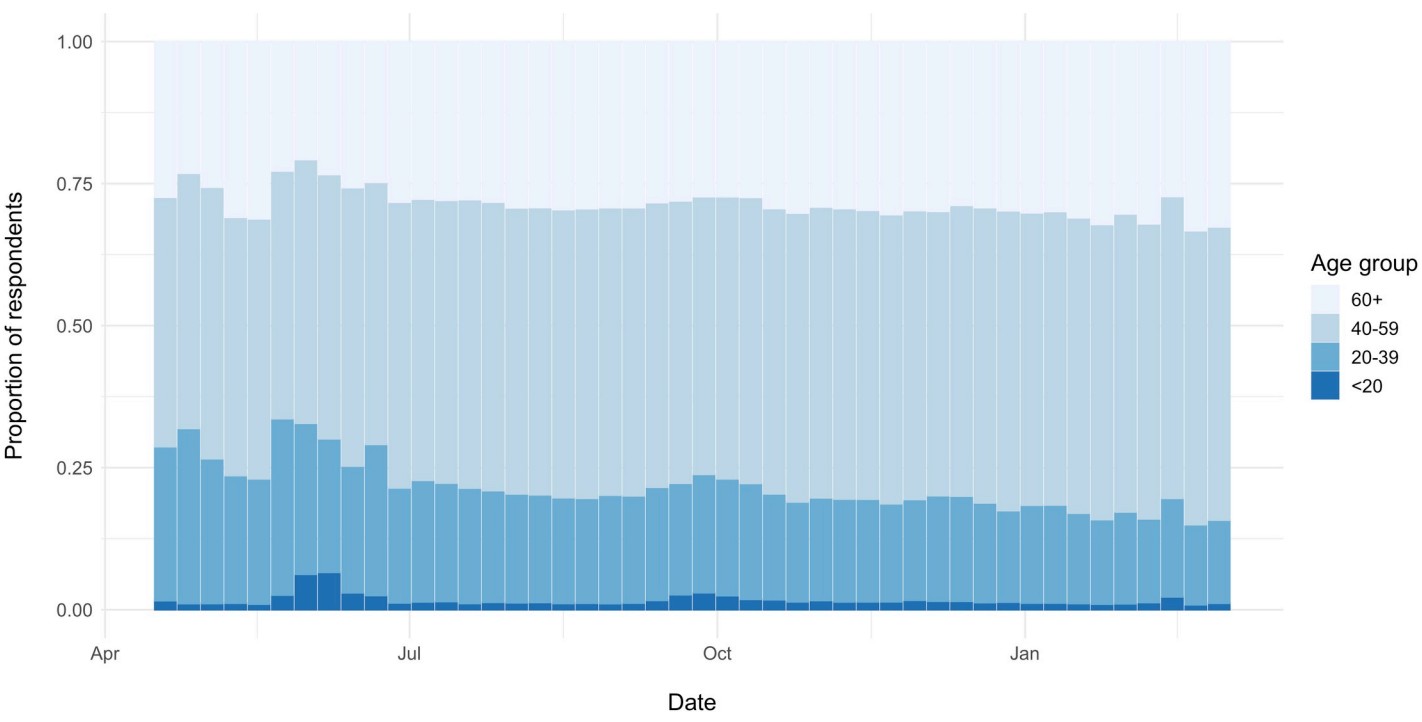

**Fig 3. Age group of ONM respondents for ISO week 17, 2020 –week 9, 2021.**

**Sex.** There was a significantly greater proportion of unique ONM respondents who identified as female (n = 41,543; 61.4% female) compared to the general Ontario population (n = 7,371,442; 50.6% female, $p < 0.01$) but less than the proportion of all Ontarians who received a test (n = 6,303,215; 63.8% female, $p < 0.01$) (Table 1). The proportion of female respondents to ONM was stable over time (Fig 1 in S1 Appendix).

**Income quintile of residential area.** There was underrepresentation of survey respondents (n = 11,388; 16.8%) living in areas in the lowest quintile of household income

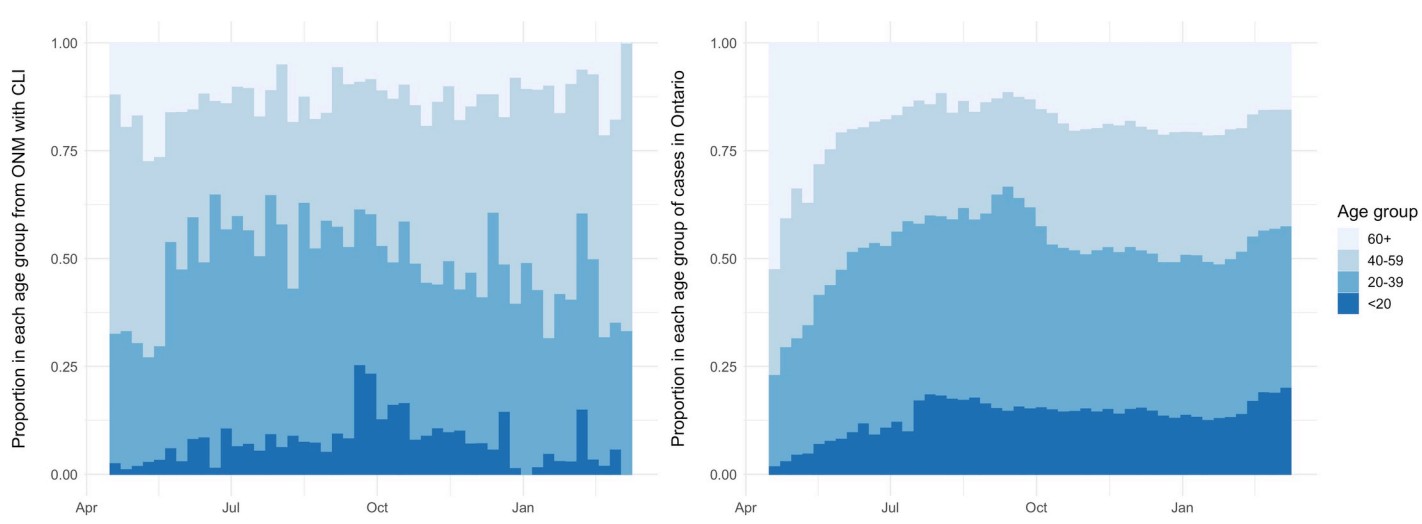

**Fig 4. Reported age of those with CLI from ONM (left) and age of reported COVID-19 cases in Ontario (right).**

**Table 1. Self-reported characteristics of respondents in data sources compared to the Ontario population.**

| | Outbreaks Near Me | Tests for COVID-19 | COVID-19 Cases | 2020 Ontario Population | Chi-Square p-value |
|---|---|---|---|---|---|
| | N = 67,693 | (N = 9,906,197) | (298,040) | N = 14,566,547 | |
| **Gender (%)** | | | | | |
| Male | 26,150 (38.6) | 3,578,181 (36.1) | 147,693 (49.9) | 7,195,105 (49.4) | $p < 0.01$[†] |
| Female | 41,543 (61.4) | 6,303,215 (63.6) | 148,758 (49.6) | 7,371,442 (50.6) | |
| Other | NA | 24,801 (0.3) | 1589 (0.5) | NA | |
| **Age group (%)** | | | | | |
| ≤19 | 3,072 (4.5) | 1,020,528 (10.3) | 41,836 (14.0) | 3,141,693 (21.6) | |
| 20–39 | 20,442 (30.2) | 2,912,608 (29.4) | 111,172 (37.3) | 4,061,469 (27.9) | |
| 40–59 | 29,206 (43.1) | 3,141,700 (31.7) | 85,804 (28.8) | 3,915,662 (26.9) | $p < 0.01$[†] |
| 60+ | 14,973 (22.1) | 2,806,560 (28.3) | 59,171 (19.9) | 3,447,723 (23.7) | |
| Not reported | NA | 24,801 (0.3) | 57 (0.02) | NA | |

[†]p-values were calculated between individuals who reported an age and gender.

(<$59,914/year) compared to COVID-19 cases (n = 65,730; 21.9%) (Table 2). The area income quintile of ONM respondents remained stable over time while the province saw fluctuations in the household income of COVID-19 cases (Fig 5). In the first wave, 50% of COVID-19 cases

**Table 2. Sociodemographic factors of outbreak Near Me respondents and COVID-19 cases in Ontario based on geographic region of dwelling.**

| | Outbreaks Near Me | Tests for COVID-19 | COVID-19 Cases | 2016 Ontario Population | Chi-Square p-value |
|---|---|---|---|---|---|
| | N = 67,693 | (9,906,197) | (N = 298,040) | (13,448,492) | |
| **Area Household Income Quintile** | | | | | |
| <59,914 | 11,388 (16.8) | 1,905,198 (19.2) | 65,439 (22.0) | 2,4008,629 (17.9) | |
| 59,914–   67,453 | 13,072 (19.3) | 2,092,497 (21.1) | 55,101 (18.5) | 2,776,337 (20.6) | |
| 67,453–   81,953 | 16,518 (24.4) | 2,240,231 (22.6) | 53,351 (17.9) | 2,912,356 (21.7) | $p < 0.01$ |
| 81,953–   98,132 | 12,478 (18.4) | 1,797,086 (18.1) | 55,532 (18.6) | 2,577,210 (19.2) | |
| >98,132 | 13,933 (20.6) | 1,817,702 (18.3) | 66,436 (22.3) | 2,773,870 (20.6) | |
| NA | 304 (0.4) | 53,483 (0.5) | 2,181 (0.7) | 90 (0.0) | |
| **Area Proportion Recent Immigrant Quintile** | | | | | $p < 0.01$ |
| <0.4% | 8,053 (11.9) | 1,733,283 (17.5) | 15,931 (5.3) | 2,185,341 (16.2) | |
| 0.4–1.1% | 8,135 (12) | 1,837,717 (18.6) | 28,562 (9.6) | 2,323,545 (17.3) | |
| 1.1–2.6% | 11,887 (17.6) | 1,803,607 (18.2) | 44,863 (15.1) | 2,404,060 (17.9) | |
| 2.6–5.3% | 18,238 (27) | 2,022,844 (20.4) | 66,293 (22.2) | 2,857,252 (21.2) | |
| 5.3+% | 21,031 (31.1) | 2,455,263 (24.8) | 140,210 (47) | 3,678,204 (27.4) | |
| NA | 304 (0.4) | 53,483 (0.5) | 2,181 (0.7) | 90 (0.0) | |
| **Area Proportion Visible Minority Quintile** | | | | | $p < 0.01$ |
| <3% | 8,190 (12.1) | 1,835,457 (18.5) | 17,215 (5.8) | 2,251,274 (16.7) | |
| 3–10% | 7,558 (11.2) | 1,831,472 (18.5) | 25,227 (8.5) | 2,449,567 (18.2) | |
| 10–23% | 12,309 (18.2) | 1,985,837 (20.0) | 42,645 (14.3) | 2,729,874 (20.3) | |
| 23–42% | 18,044 (26.6) | 1,665,802 (16.8) | 60,111 (20.2) | 2,141,107 (15.9) | |
| >42% | 21,288 (31.4) | 2,534,146 (25.6) | 151,661 (50.6) | 3,876,480 (28.8) | |
| NA | 304 (0.4) | 53,483 (0.5) | 2,181 (0.7) | 90 (0.0) | |
| **Type of dwelling area** | | | | | |
| Rural Area | 6,756 (10) | 1,389,431 (14.0) | 16,793 (5.6) | 1,848,110 (13.7) | $p < 0.01$ |
| Urban Area | 60,650 (89.6) | 8,482,671 (85.6) | 280,208 (94.0) | 11,600,382 (86.3) | |
| NA | 287 (0.42) | 34,095 (0.3) | 1039 (0.3) | NA | |

Bins represent the 5 quintiles of the Ontario population

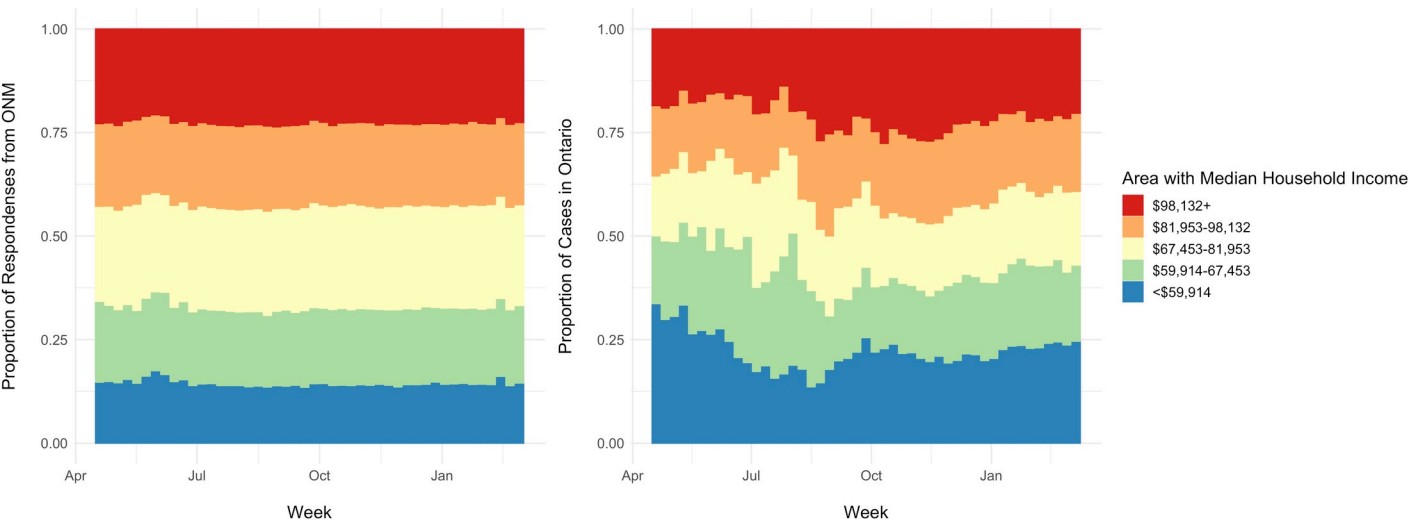

**Fig 5. Household income in ONM and COVID-19 cases.** Distribution of responses in each quintile from ONM and COVID-19 cases in Ontario over time based on median annual household income (Canadian Dollars) in geographic area.

came from areas in the lowest two quintiles of annual household income (April 2020). This trend was not seen in ONM responses (Fig 5).

**Immigration quintile of residential area.** Marked differences were seen between the immigration quintiles of the residential areas of ONM respondents and COVID-19 cases. Cases in Ontario overrepresented areas with the highest quintile of recent immigrants (>5.3% recent immigrants). There were 140,687 (47%) cases living in areas with >5.3% recent immigrants. In contrast, only 21,031 (31.1%) respondents lived in areas with >5.3% recent immigrants (Table 2) Over time, the highest proportion of COVID-19 cases consistently came from geographic areas in the highest recent immigrant quintile. This observation was not seen in ONM respondents (Fig 6).

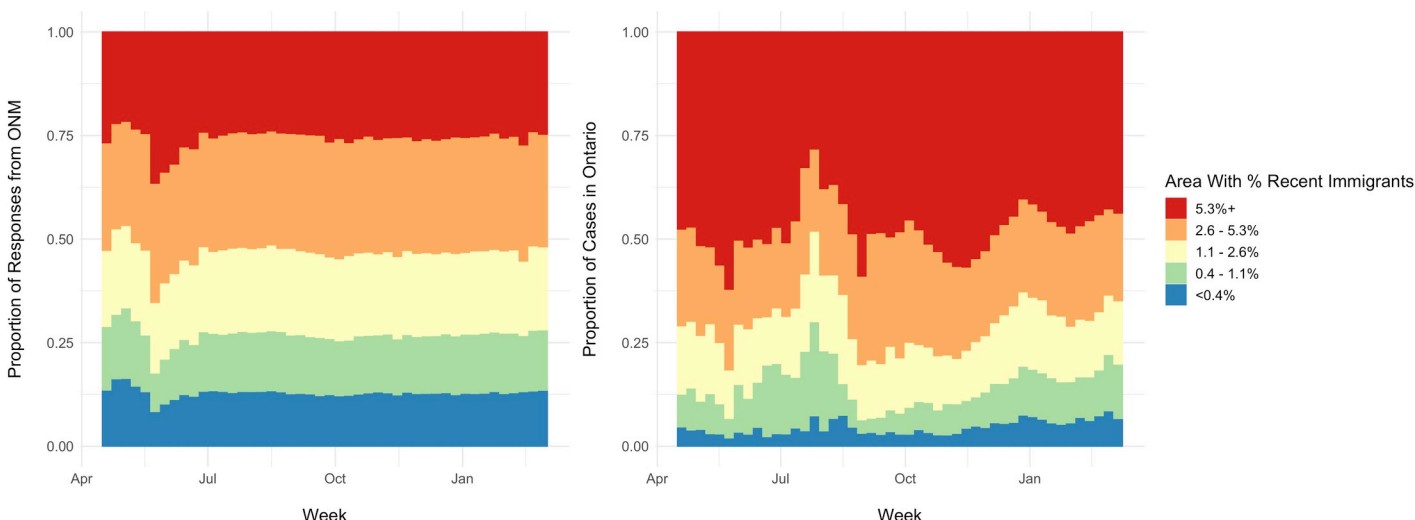

**Fig 6. Percent recent immigrants in ONM and COVID-19 cases.** Distribution of responses in each quintile from ONM and COVID-19 cases in Ontario over time based on proportion recent immigrants (last 5 years) in geographic area sorted by quintile.

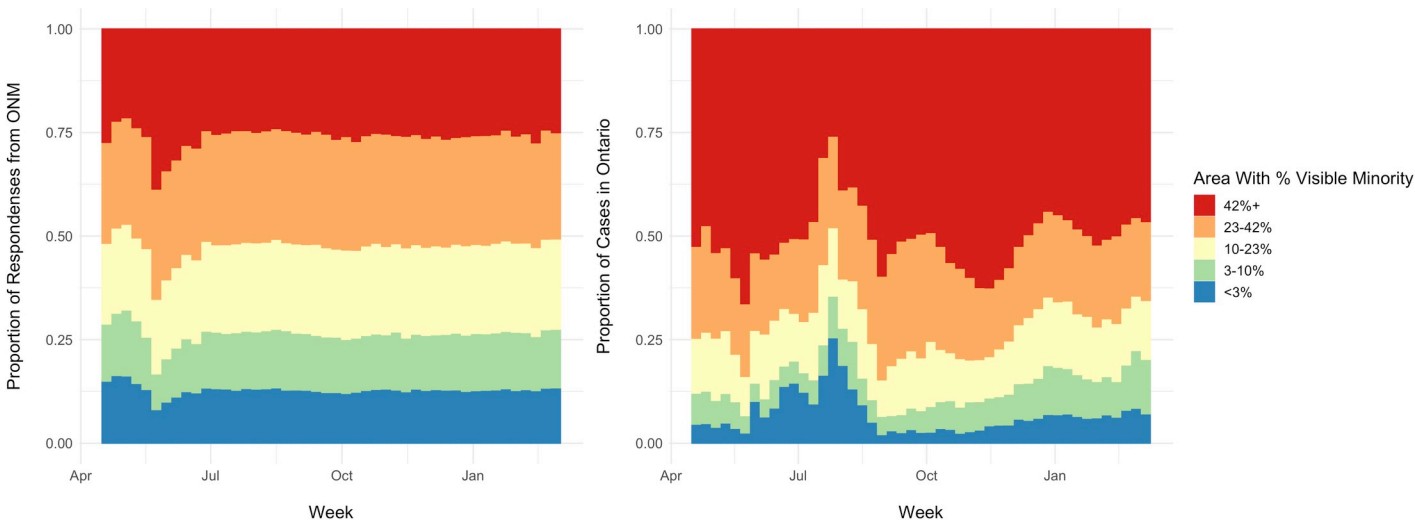

**Fig 7. Percent visible minorities in ONM and COVID-19.** Distribution of responses in each quintile from ONM and COVID-19 cases in Ontario over time based on % visible minorities in geographic area.

**Visible minority quintile of residential area.**     Large differences were also seen between the visible minority quintiles of the residential areas of ONM respondents and COVID-19 cases. Cases in Ontario were heavily overrepresented in individuals from areas with the highest quintile of visible minorities. There were 151,117 (50.5%) cases living in areas with >42% visible minorities. This was significantly lower in ONM with 21,288 (31.4%) respondents living in areas with >42% visible minorities (Table 2). Over time, the highest proportion of COVID-19 cases consistently came from geographic areas with the highest quintile of percent visible minorities. This observation was not seen in ONM respondents (Fig 7).

**Rurality of residential area.**     ONM respondents were slightly enriched in those that came from urban areas (89.6%) compared to that of the Ontario Population (86.3% urban dwelling). However, COVID-19 cases in Ontario were more heavily localized to urban areas (94.0% of cases) than ONM respondents (89.6%) (Table 2).

## Discussion

We found that there was no correlation between self-reported COVID-like illness (CLI) and the number of new COVID-19 reported cases or weekly COVID-19 precent positivity during the period of April 2020 –March 2021 in Ontario. We previously reported that the CLI definition tracked with rhinovirus and enterovirus in fall 2020 in Ontario, likely due to syndromic overlap [16]. Although syndromic definitions were correlated with COVID-19 case counts prior to the rise in rhinovirus/enterovirus in fall in 2020, this has not been the case in winter-spring 2021. Even after the weeks with high rhinovirus positivity, we observed no consistent correlation between symptom trends and COVID-19 case counts (Table 1 in S1 Appendix). Yet, syndromic reports correlated well across data sources (ONM and FluWatchers). This lack of correlation between syndromic data and confirmed cases counts was seen among 3 different syndromic definitions (Table 1 in S1 Appendix). All syndromic definitions showed high correlation with confirmed cases before the spike in Rhinovirus but also tracked with Ontario rhinovirus spike. Even the CLI$_3$ definition with 95% specificity to confirmed SARS-CoV2 was affected by rhinovirus, likely indicating heavy syndromic overlap between the two respiratory illnesses. Further it was seen that nearly all symptoms tracked with each other–all showing a

spike during the rhinovirus rise in late summer in Ontario (Fig 3 in S1 Appendix). This indicates it is unlikely that any combination of symptoms would have been unaffected by the rhinovirus peak in fall 2020.

We did observe strong positive correlation between those reporting close contact with a confirmed COVID-19 case and the province-wide count of confirmed cases. This was an expected result as the probability of having a close contact is expected to rise with the known burden of COVID-19 at any given time. Adding self-reported CLI to close contact status did not improve the correlation with province-wide cases; in fact, it fell slightly. Unlike purely syndromic definitions, awareness of being a close contact depends on cases having access to testing. As one aim of syndromic surveillance is to identify trends before they are detected through testing, this would be a limitation of such an approach in times were testing is less accessible, such as at the onset of a pandemic.

Yoneoka et al. and Nomura et al. reported analyses of syndromic data collected through a large-scale (over 350,000 participants) digital surveillance system in Tokyo, Japan. Strong spatial correlations were seen between syndromic data and COVID-19 during one week in the first wave of the Japanese COVID-19 endemic. We also found positive longitudinal correlation between CLI and various COVID-19 metrics in Ontario early in the pandemic. Over the course of Ontario's endemic, we found no correlation between COVID-19 activity and self-reported COVID-like illness. The characteristics of respondents to ONM remained similar over time (Figs 3 and 5–7 and Fig 1 in S1 Appendix) indicating a relatively consistent cohort of weekly respondents. It is possible that symptoms of COVID-19 may have been present and detected in a fraction of higher-risk individuals in this cohort early in the pandemic, but that these same individuals become less susceptible over successive waves, due to immunity or high levels of health consciousness and related cautious behaviour.

We found significant differences in age, gender and residential area income level, proportion of visible minorities, and proportion of recent immigrants. ONM respondents were more likely to be female and aged 40–59 years than those being tested for SARS-CoV2 in Ontario. Others have similarly reported that middle-aged females were the group most engaged with influenza participatory surveillance tools [6]. Yet, in Ontario, approximately 50% of COVID-19 cases were being reported by those 60+ in April 2020. As the province saw large volumes of cases localized to long-term care homes and retirement residences in the first and second waves, this could be one explanation for the relative undercounting of COVID-19 disease activity among older age groups by self-reported symptoms data [29].

In addition, Ontario's COVID-19 cases came disproportionately from areas in the lowest income quintile, and the highest quintile of recent immigrants and visible minorities. ONM participatory surveillance method relies on access to the internet, which may exclude individuals who are underhoused or experiencing homelessness, those with poor internet or computer access, or limited English literacy. These characteristics are more common among the low income and marginalized groups who were disproportionately affected by COVID-19 [30].

A strength of this study includes the use of four separate syndromic definitions over a range of varying sensitivities and specificities for confirmed SARS-CoV2. We used three independent CLI definitions and an ILI syndromic definition. Longitudinal trends were similar across all syndromic definitions. A strength of the ONM tool is the longitudinal retention of a large proportion of survey respondents through text reminders, reducing the risk of inflated symptom estimates resulting from response bias. A limitation of our demographic analysis of survey respondents is that we do not have individual-level information on income, proportion of visible minorities or recent immigrants. Forward sortation areas are much larger than individual neighborhoods and ecological bias is possible. Nonetheless, our findings are consistent with

those of others who found a higher proportion of affluent and educated long-term respondents to participatory surveillance tools for influenza [6].

## Conclusion

Participatory surveillance tools have demonstrated utility in the early identification of influenza outbreaks, as well as geospatial identification of COVID-19 outbreaks. We found that, despite good uptake, a participatory surveillance tool showed poor longitudinal correspondence with COVID-19 case counts in Ontario, Canada. Self-reported close contact with a COVID-19 case did show a strong association with case activity in the province. We also found discrepancies between participatory surveillance respondents and the Ontario population in income and the proportion of immigrants, visible minorities and those living in rural areas. This is the first long-term comparison of participatory surveillance data to COVID-19 case activity. Although digital surveillance systems such as ONM are low-cost tools that may be helpful in determining the burden of COVID-19 in certain regions, various factors such as seasonal respiratory virus transmission, a consistent cohort of respondents, and differing population coverage may limit correspondence with longitudinal trends in confirmed COVID-19 case activity.

## Supporting information

**S1 Appendix.**
(DOCX)

**S1 Dataset. Aggregate data.**
(XLSX)

## Acknowledgments

The authors have no specific acknowledgements.

## Author Contributions

**Conceptualization:** Isaac I. Bogoch, Christina M. Astley, David N. Fisman, John S. Brownstein, Lauren Lapointe-Shaw.

**Data curation:** Arjuna S. Maharaj, Jennifer Parker, Benjamin Rader, Jared B. Hawkins, Liza Lee.

**Formal analysis:** Arjuna S. Maharaj, Jennifer Parker, Benjamin Rader, Ashleigh R. Tuite, Lauren Lapointe-Shaw.

**Methodology:** Arjuna S. Maharaj, Jessica P. Hopkins, Effie Gournis, Christina M. Astley, David N. Fisman, John S. Brownstein, Lauren Lapointe-Shaw.

**Software:** Jared B. Hawkins.

**Visualization:** Arjuna S. Maharaj, Ashleigh R. Tuite.

**Writing – original draft:** Arjuna S. Maharaj, Lauren Lapointe-Shaw.

**Writing – review & editing:** Jennifer Parker, Jessica P. Hopkins, Effie Gournis, Isaac I. Bogoch, Benjamin Rader, Christina M. Astley, Noah M. Ivers, Jared B. Hawkins, Liza Lee, Ashleigh R. Tuite, David N. Fisman, John S. Brownstein, Lauren Lapointe-Shaw.

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
