## [Decision Letter · Decision Letter 0]

5 Oct 2021

PONE-D-21-20494Comparison of longitudinal
trends in self-reported symptoms and COVID-19 case activity in Ontario,
CanadaPLOS ONE

Dear Dr. Maharaj,

Thank you for submitting your manuscript to PLOS ONE. After careful consideration, we
feel that it has merit but does not fully meet PLOS ONE’s publication criteria as it
currently stands. Therefore, we invite you to submit a revised version of the
manuscript that addresses the points raised during the review process.

The Authors are expected to address all the criticisms by all Reviewers. In
particular, please reconsider the use of symptoms only approach for the surveillance
of COVID-19, and the conclusion on the use of participatory surveillance or
specifically the adopted CLI definition (Reviewer #1) and strengthen the discussion
(Reviewer #2). In additional to the above comments, please address,

To fully assess the use of participatory surveillance using a symptom
only approach, the authors may consider other combinations symptom which
may be less affected by other respiratory infections (e.g. rhinoviruses
or enteroviruses). Such alternative definitions have been considered in
Reses et al. (BMC Public Health, 2021). Please submit your revised manuscript by Nov 19 2021 11:59PM. If you will
need more time than this to complete your revisions, please reply to this message or
contact the journal office at plosone@plos.org.
When you're ready to submit your revision, log on to https://www.editorialmanager.com/pone/ and select the 'Submissions
Needing Revision' folder to locate your manuscript file.

Please include the following items when submitting your revised
manuscript:A rebuttal letter that responds to each point raised by the academic
editor and reviewer(s). You should upload this letter as a separate file
labeled 'Response to Reviewers'.A marked-up copy of your manuscript that highlights changes made to the
original version. You should upload this as a separate file labeled
'Revised Manuscript with Track Changes'.An unmarked version of your revised paper without tracked changes. You
should upload this as a separate file labeled 'Manuscript'.

If you would like to make changes to your financial disclosure, please include your
updated statement in your cover letter. Guidelines for resubmitting your figure
files are available below the reviewer comments at the end of this letter.

We look forward to receiving your revised manuscript.

Kind regards,

Eric HY Lau, Ph.D.

Academic Editor

PLOS ONE

Additional Editor Comments (if provided):

The Authors are expected to address all the criticisms by all Reviewers. In
particular, please reconsider the use of symptoms only approach for the surveillance
of COVID-19, and the conclusion on the use of participatory surveillance or
specifically the adopted CLI definition (Reviewer #1) and strengthen the discussion
(Reviewer #2). In additional to the above comments, please address,

1. To fully assess the use of participatory surveillance using a symptom only
approach, the authors may consider other combinations symptom which may be less
affected by other respiratory infections (e.g. rhinoviruses or enteroviruses). Such
alternative definitions have been considered in Reses et al. (BMC Public Health,
2021).

3. Thank you for stating the following in the Competing Interests/Financial
Disclosure * (delete as necessary) section:

“I have read the journal's policy and the authors of this manuscript have the
following competing interests:

IIB has consulted to BlueDot, a social benefit corporation that tracks the spread of
emerging infectious diseases.

DNF reports personal fees from Pfizer, AstraZeneca, and Seqirus, outside the
submitted work.”

We note that you received funding from a commercial source: “Pfizer, AstraZeneca, and
Seqirus”

Please include your amended Competing Interests Statement within your cover letter.
We will change the online submission form on your behalf

Reviewers' comments:

Reviewer's Responses to Questions

**Comments to the Author**

1. Is the manuscript technically sound, and do the data support the conclusions?

Reviewer #1: No

Reviewer #2: Partly

2. Has the statistical analysis been performed
appropriately and rigorously? 

Reviewer #1: Yes

Reviewer #2: Yes

3. Have the authors made all data underlying the
findings in their manuscript fully available?

Reviewer #1: No

Reviewer #2: No

4. Is the manuscript presented in an intelligible
fashion and written in standard English?

Reviewer #1: Yes

Reviewer #2: Yes

5. Review Comments to the Author

Reviewer #1: The authors compared longitudinal trends in self-reported symptoms and
COVID-19 case activity in Ontario, Canada, and concluded that a participatory
surveillance tool showed poor longitudinal correspondence with COVID-19 case counts.
The conclusion doesn’t seem reliable with current data analysis.

Major comments:

1. The major issue comes from definition of COVID-like illness (CLI), defined by

“the presence of at least two of: fever (measured or subjective), chills, ... or new
taste disorder. “ Though they cited these symptoms from CDC website (https://ndc.services.cdc.gov/case-definitions/coronavirus-disease-2019-2021/),
the CDC has never defined these symptoms as CLI. Indeed, on the same webpage, under
the section of “Case Classification > Probable”, the CDC requires a case to be
probable, under the condition when no confirmatory or presumptive laboratory
evidence for SARS-CoV-2 is available, to have at least “epidemiologic linkage” which
is not reported in current study BUT can surely be obtained using a participatory
approach, as shown in other published studies. Thus, the conclusion can only be that
using the symptom-only survey cannot help surveil COVID-19. By no mean can it be
concluded that the participatory surveillance tool does not work. In fact, the
participatory surveillance works very well with the large amount of data collected
in a short period of time, as shown by the authors.

Reviewer #2: The manuscript is well written. I believe it is worth to be published in
Plos One. I however suggest that the authors move the historical COVID 19 trends in
Ontario to the background section. The authors also refer to a 2016 census in the
methodology section but do not provide a citation for it. The authors have
rigorously presented the findings but sort of gross over them in the discussion
section. Strengthening of the discussion section would further enrich the
manuscript. Lastly, the conclusion in the main document is a bit light weight
considering the amount of findings presented.

6. PLOS authors have the option to publish the peer
review history of their article (what does this mean?). If published, this will
include your full peer review and any attached files.

If you choose “no”, your identity will remain anonymous but your review may still be
made public.

**Do you want your identity to be public for this peer review?** For
information about this choice, including consent withdrawal, please see our
Privacy Policy.

Reviewer #1: No

Reviewer #2: No

---

## [Author Response · Author response to Decision Letter 0]

2 Nov 2021

Reviewer #1 

1.) Definition of CLI

Thank you for this excellent point on including epidemiologic linkage which was not
reported in our original submission. Indeed, this was available in the Outbreaks
Near Me tool. We have added an additional analysis to our manuscript where we
compare the proportion of weekly participatory surveillance respondents who reported
close contact to confirmed COVID-19 case to weekly cases in Ontario. This was also
combined this with symptom data such that those with both CLI and contact were
compared to weekly cases. This resulted in a new CLI definition as suggested: CLI +
epidemiologic linkage.

We found that there was a strong association between the proportion of those
reporting close contact and cases in Ontario (ρ = 0.77). There was also a strong
association between those with contact + CLI symptoms and cases in Ontario (ρ =
0.70). Close contact did track with the second wave in Ontario while symptom data
alone did not. These were expected findings as self-reported close contact reflects
burden of disease in an area. This finding has been added to Supplementary figure 6
with integration into the discussion and conclusion section. 

Reviewer #2

1.) Move Historical Trends of COVID-19 to backgrounds section

Thank you for this recommendation. Our section on historical trends has now been
moved to the background section

2.) Missing citation for 2016 census methodology

We apologize for this oversight. A citation has been added for the 2016 census
methodologies

3.) Strengthening discussion

Thank you for this recommendation. We have added additional points to our discussion
including further commentary on similarities in trends seen between different
syndromic definitions, commentary on adding epidemiologic linkage to our syndromic
definitions, and additions to the strengths section of the manuscript. 

4.) Conclusion Section 

We have added two additional conclusion points namely on our new finding of
self-reported direct contact and integrating our finding on differences between
participatory respondent demographic and Ontario population demographics. 

Editor comments 

1.) Other combinations symptom which may be less affected by other respiratory
infections

Thank you for this suggestion. We agree that testing other combinations of symptoms
are important to uncover one that may not be affected by other infections. We have
reviewed Reses et al. in depth and have chosen a symptom combination with the
highest specificity for COVID-19 that was also available with the symptoms surveyed
by ONM. This was added to our paper as CLI3 and consisted of taste and/or smell
dysfunction, or one of the following: shortness of breath, myalgia, or fever or
chills. This was listed as derived compound combination 1 in Reses et al and added
to Supplement Figure 5. We found that this symptom combination also did not
correlate with weekly cases and in fact also followed the Rhinovirus spike. We
believe it to be a strength of our paper to now have 4 syndromic definitions. All 4
showed very similar trends over time. Further, we included a breakdown of all
symptom data over time in Supplement Figure 3. Here we see all symptom components
track with each other (all experiencing a rhinovirus spike) indicating that it is
unlikely that a specific combination of symptoms would track only with COVID-19 in
Ontario. This indicates the heavy syndromic overlap between COVID-19 and other
respiratory viruses.

to reviewers.docx
---

## [Decision Letter · Decision Letter 1]

26 Dec 2021

Comparison of longitudinal trends in self-reported symptoms and COVID-19 case
activity in Ontario, Canada

PONE-D-21-20494R1

Dear Dr. Maharaj,

We’re pleased to inform you that your manuscript has been judged scientifically
suitable for publication and will be formally accepted for publication once it meets
all outstanding technical requirements.

Kind regards,

Eric HY Lau, Ph.D.

Academic Editor

PLOS ONE

Additional Editor Comments (optional):

Reviewers' comments:

Reviewer's Responses to Questions

**Comments to the Author**

1. If the authors have adequately addressed your comments raised in a previous round
of review and you feel that this manuscript is now acceptable for publication, you
may indicate that here to bypass the “Comments to the Author” section, enter your
conflict of interest statement in the “Confidential to Editor” section, and submit
your "Accept" recommendation.

Reviewer #2: All comments have been addressed

2. Is the manuscript technically sound, and do the data
support the conclusions?

Reviewer #2: Yes

3. Has the statistical analysis been performed
appropriately and rigorously? 

Reviewer #2: Yes

4. Have the authors made all data underlying the
findings in their manuscript fully available?

Reviewer #2: No

5. Is the manuscript presented in an intelligible
fashion and written in standard English?

Reviewer #2: Yes

6. Review Comments to the Author

Reviewer #2: Thank you for addressing my comments. Minor comment: please insert links
to the web pages referred to in the document.

7. PLOS authors have the option to publish the peer
review history of their article (what does this mean?). If published, this will
include your full peer review and any attached files.

If you choose “no”, your identity will remain anonymous but your review may still be
made public.

**Do you want your identity to be public for this peer review?** For
information about this choice, including consent withdrawal, please see our
Privacy Policy.

Reviewer #2: No

---

## [Editor Report · Acceptance letter]

3 Jan 2022

PONE-D-21-20494R1 

Comparison of longitudinal trends in self-reported symptoms and COVID-19 case
activity in Ontario, Canada 

Dear Dr. Maharaj:

I'm pleased to inform you that your manuscript has been deemed suitable for
publication in PLOS ONE. Congratulations! Your manuscript is now with our production
department. 

Kind regards, 

on behalf of

Dr. Eric HY Lau 

Academic Editor

PLOS ONE